# PKN1 Exerts Neurodegenerative Effects in an In Vitro Model of Cerebellar Hypoxic–Ischemic Encephalopathy via Inhibition of AKT/GSK3β Signaling

**DOI:** 10.3390/biom13111599

**Published:** 2023-10-31

**Authors:** Stephanie zur Nedden, Motahareh Solina Safari, Friedrich Fresser, Klaus Faserl, Herbert Lindner, Bettina Sarg, Gottfried Baier, Gabriele Baier-Bitterlich

**Affiliations:** 1Institute of Neurobiochemistry, CCB-Biocenter, Medical University of Innsbruck, 6020 Innsbruck, Austria; motahareh.safari@i-med.ac.at; 2Institute for Cell Genetics, Medical University of Innsbruck, 6020 Innsbruck, Austria; friedrich.fresser@i-med.ac.at (F.F.); gottfried.baier@i-med.ac.at (G.B.); 3Protein Core Facility, Institute of Medical Biochemistry, CCB-Biocenter, Medical University of Innsbruck, 6020 Innsbruck, Austria; klaus.faserl@i-med.ac.at (K.F.); herbert.lindner@i-med.ac.at (H.L.); bettina.sarg@i-med.ac.at (B.S.)

**Keywords:** hypoxic–ischemic encephalopathy, cerebellum, protein kinase N1, AKT

## Abstract

We recently identified protein kinase N1 (PKN1) as a negative gatekeeper of neuronal AKT protein kinase activity during postnatal cerebellar development. The developing cerebellum is specifically vulnerable to hypoxia-ischemia (HI), as it occurs during hypoxic-ischemic encephalopathy, a condition typically caused by oxygen deprivation during or shortly after birth. In that context, activation of the AKT cell survival pathway has emerged as a promising new target for neuroprotective interventions. Here, we investigated the role of PKN1 in an in vitro model of HI, using postnatal cerebellar granule cells (Cgc) derived from *Pkn1* wildtype and *Pkn1^−/−^* mice. *Pkn1^−/−^* Cgc showed significantly higher AKT phosphorylation, resulting in reduced caspase-3 activation and improved survival after HI. *Pkn1^−/−^* Cgc also showed enhanced axonal outgrowth on growth-inhibitory glial scar substrates, further pointing towards a protective phenotype of *Pkn1* knockout after HI. The specific PKN1 phosphorylation site S374 was functionally relevant for the enhanced axonal outgrowth and AKT interaction. Additionally, PKN1pS374 shows a steep decrease during cerebellar development. In summary, we demonstrate the pathological relevance of the PKN1-AKT interaction in an in vitro HI model and establish the relevant PKN1 phosphorylation sites, contributing important information towards the development of specific PKN1 inhibitors.

## 1. Introduction

Hypoxic–ischemic encephalopathy (HIE) is the leading cause of death and disability in newborn humans, often resulting in persistent motor, sensory and cognitive impairment. Recently, it was suggested that the cerebellum may be involved in neonatal HIE since high rates of cerebellar injury can be visualized using neuropathology and advanced MRI tools [1,2], and cerebellar vermian atrophy is frequently seen in patients with neonatal HIE and thalamic edema [3]. Many pathways, such as excitotoxicity, oxidative stress and inflammation, have been implicated in HIE brain injury, and molecular studies have focused attention on the development of new therapies targeting those processes [4,5]. Furthermore, HIE brain injury is characterized by glial scarring [6,7], which prevents regenerative attempts of CNS axons [8]. Consequently, interventions that target both neuronal cell survival and regenerative capacity could have enormous therapeutic potential in HIE. Therefore, a greater understanding of the molecular mechanisms involved in regulating hypoxia-ischemia (HI)-induced neuronal cell death and axonal regeneration in the developing brain will contribute to advancing the development of therapeutic interventions [9].

Our group focuses on the critical function of the protein kinase N1 (PKN1) in the brain. PKN1 is a serine/threonine kinase and belongs to the PKC superfamily, sharing a characteristic C-terminal catalytic domain [10,11] that requires phosphorylation by phosphoinositide-dependent kinase-1 (PDK1) for activation. The N-terminal regulatory domain confers binding and regulation by RhoA/B/C, Rac1 [12,13,14], fatty acids, and phospholipids [15,16]. The cerebellum, which is central to motor control and coordination, is also the brain area with the highest levels of PKN1 expression [17]. We found that PKN1 acts as a developmentally active gatekeeper of AKT activity and thereby controls cerebellar synapse formation and long-term function [18]. We also reported that the hippocampus shows a similar dysregulation of AKT upon *Pkn1* knockout, suggesting that the tight control of postnatal AKT levels by PKN1 constitutes a general regulatory mechanism in the postnatal development of various brain regions [19].

In recent years, the activation of the phosphoinositide 3-kinase/AKT cell survival pathway has emerged as a promising new target for neuroprotective interventions [20,21]. Since AKT is a central protein kinase in neuroprotection and neuroregeneration, such as spinal cord injury [22], stroke [21], and HIE [23], we speculated that the PKN1-AKT interaction might also be relevant during pathophysiological conditions during development. We tested this hypothesis in an in vitro neonatal HI model using postnatal murine Cgc, prepared from wildtype or *Pkn1^−/−^* mice [24]. We show that PKN1-mediated AKT inhibition contributes to HI-induced cell death and further prevents axonal outgrowth on growth-inhibitory substrates of the glial scar. We further describe the phosphorylation sites of PKN1 that are functionally important for the regulation of axonal outgrowth and AKT interaction.

## 2. Materials and Methods

### 2.1. Animals

The generation of *Pkn1* knockout mice (*Pkn1*^−/−^ mice) has been described recently [24]. Animals were kindly provided by P. Parker (Francis Crick Institute of London, UK) and A. Cameron (Queen Mary University of London, UK). Mice were backcrossed to C57BL/6N for more than 10 generations. C57BL/6N wildtype (WT) and C57BL/6N *Pkn1^−/−^* animals were derived from the same heterozygous crosses and then bred separately but kept under the same housing and experimental conditions in the same room. C57BL/6N were derived from The Jackson Laboratory. All animals in this study (younger than postnatal day (P)12) were killed by decapitation.

### 2.2. Preparation of Cerebellar Granule Cells

Cgc were prepared from 6 to 8-day-old mouse pups, as described previously [25]. Cells were kept in a supplemented Neurobasal medium (Thermo Fisher Scientific, Waltham, MA, USA), 1% Penicillin/Streptomycin/Glutamine, 2% B-27, 20 mM KCl). Coverslips and dishes were either coated with Poly-L-Ornithine (2–3 h, Sigma) or Poly-L-Lysin (2 h-overnight, Sigma (St. Louis, MO, USA) and subsequently with chondroitin sulfate proteoglycans (CSPG, 1–10 µg/mL, Merck Millipore, Darmstadt, Germany), a combination of laminin (Sigma)/CSPG (5 µg/mL each) or laminin (10 µg/mL) for 2–3 h at 37 °C. Cgc were kept in a humidified incubator at 37 °C for indicated timepoints.

### 2.3. Preparation of Plasmids and Transfections of Cerebellar Granule Cells

Human (*h*) PKN1 was purchased from GeneCopoeia (Rockville, MD, USA) and subcloned into the mammalian expression vector pTO_HA_StrepIII_c_GW_FRT_EF1alpha. The kinase mutants S374A, TT63/64AA, and T744A were generated by site-directed mutagenesis.

Nucleofection of *Pkn1^−/−^* Cgc was performed as described previously [18], with 5 μg wildtype or mutant *h*PKN1 together with 3 μg pmax GFP plasmid (Lonza, Basel, Switzerland) using the program G-013 (Lonza). Cgc were kept for 1 day in vitro (DIV1) on Laminin-coated coverslips, fixed with PFA and GFP-positive cells were analyzed for axonal length and outgrowth as described below. For Lipofectamine transfections (Life Technologies, Thermo Fisher (Waltham, MA, USA)), 2 μL Lipofectamine and 0.9 μg plasmid DNA were mixed with Neurobasal medium and added to *Pkn1^−/−^* Cgc coverslips after DIV5 for 6–8 h. The medium was then exchanged for the preconditioned culture medium, and cells were analyzed after 48 h.

### 2.4. Immunofluorescence Staining

Cgc were fixed (4% PFA 15 min, methanol 30 s −20 °C), permeabilized (0.3% Triton-X-100, 15 min), and blocked (2% BSA, 10% goat serum, 1 h). The primary antibodies (diluted in 0.1% Triton-X-100, 1% BSA, and 5% goat serum in PBS) were added at 4 °C overnight. After washing in PBS, secondary antibodies (goat-anti rabbit Alexa-488 A11070, and goat anti-mouse Alexa-555 A21425, Molecular Probes, Life technologies), as well as Hoechst (8 µM, Molecular Probes, Life technologies), were added for 2–4 h at room temperature. Coverslips were washed in PBS and embedded in Mowiol (Sigma). Images were taken with a widefield (Axio, Zeiss, Oberkochen, Germany) microscope. Jurkat suspension cells were harvested and centrifuged (2 min, 4000 rpm at RT). The pellets were washed in PBS, centrifuged again, and resuspended in 4% PFA for 13 min at room temperature. Following fixation, the cells were permeabilized (0.3% Triton-X-100, 8 min) and blocked (2% BSA, 10% goat serum, 18 min). The primary antibodies (diluted in 0.3% Triton-X-100, 1% BSA, and 5% goat serum in PBS) were added and incubated for 3 h at room temperature. After washing in PBS, secondary antibodies and Hoechst were added for 45 min at room temperature. Cells were washed in PBS, centrifuged (2 min, 4000 rpm), and the pellets were subsequently resuspended in 8 µL of Mowiol (Sigma). Images were taken with a confocal (LSM980 with Airyscan 2, Zeiss) microscope.

### 2.5. Measurement of Axonal Length

Cells, grown on CSPG- or Laminin/CSPG-coated coverslips were fixed at DIV1, stained for TAU, and analyzed for axonal length as described previously [18]. Pictures were manually cleaned from background noise using ImageJ. Incomplete neurons, as well as neurons with axons that crossed other axons (in that case, the neuron with the shorter axon, or the one that crossed more than two other axons was erased), were erased. A minimum of 70 cells/coverslip (DIV1) were analyzed with WIS Neuromath [26,27]. For wildtype or mutant *h*PKN1-transfected *Pkn1^−/−^* Cgc, cells were stained with TAU. GFP-positive (transfected) or negative (untransfected) cells were imaged on the same coverslip and analyzed as described above. Axonal length of transfected cells is expressed as a percentage of axonal length of untransfected cells.

### 2.6. HI-Model

After 6-8 DIV Cgc were exposed to HI by replacing the culture medium with glucose-free DMEM (equilibrated with 95% N_2_ and 5% CO_2_ overnight, 2% B-27, 1% Penicillin/Streptomycin/Glutamine) and keeping the cells in an airtight chamber (Billups-Rothenberg Inc., San Diego, CA, USA)) purged with 95% N_2_ and 5% CO_2_ for 3 h (oxygen–glucose deprivation, OGD). The medium of control cells was replaced with glucose-containing supplemented DMEM. Reperfusion (Rep) was achieved by adding the preconditioned culture medium back to the cells for indicated timepoints.

### 2.7. Measurement of pAKT Intensity in Cerebellar Granule Cells

*Pkn1^−/−^* Cgc were Lipofectamine-transfected with wildtype *h*PKN1 on DIV5. At DIV7, Cgc were exposed to 3 h OGD and 1 h Rep, fixed and stained for HA (to visualize *h*PKN1) and pAKT T308 (Sigma). Images were taken with a widefield (Axio, Zeiss) microscope, and mean pAKT levels were measured with ImageJ, version 1.54g (raw intensity/area) in the transfected (HA-positive) cell and the surrounding untransfected cells for each condition (control or HI). The mean pAKT intensity of 6–26 transfected cells and of the surrounding 1–10 untransfected cells was analyzed from 1 to 3 coverslips/group and preparation. The mean pAKT intensity in transfected cells is expressed as a percentage of untransfected cells for each condition (control or HI).

### 2.8. Western Blotting

Protein extracts of whole cerebella from postnatal day 1 (P1)–P8 were prepared (50 mM Tris pH 8.5, 1% NP-40, 5 mM EDTA, 5 mM sodium pyrophosphate, 5 mM sodium fluoride, 50 mM sodium chloride, 5 mM activated sodium orthovanadate, 30 µg/mL aprotinin, 30 µg/mL leupeptin) and analyzed by western blotting as described previously [28]. Protein extracts from Jurkat cells were prepared in a high salt extraction buffer (50 mM Tris pH 7.3, 1% NP-40, 5 mM EDTA, 5 mM sodium pyrophosphate, 5 mM sodium fluoride, 150 mM sodium chloride, 5 mM activated sodium orthovanadate, 50 µg/mL aprotinin, 50 µg/mL leupeptin) and analyzed by western blotting as described previously [29]. Primary antibodies were added over night in 5% BSA in TBS-T at 4 °C and secondary antibodies (LiCor, anti-mouse 680 nm, shown in red and anti-rabbit 800 nm, shown in green) or HRP-tagged antibodies were added for 90 min in 5% Milk in TBST. After washing in TBS-T membranes were imaged and analyzed with Odysee infrared Imager (LiCor, Bad Homburg, Germany) or ECL detection as described previously [30].

### 2.9. Antibodies

All catalogue numbers and dilutions of all antibodies used throughout the study are listed below in Table 1.

### 2.10. Protein Kinase Assay

PKN1-dependent phosphorylation was measured by incorporation of inorganic phosphate ^32^Pi from γ-^32^P-ATP. Purified recombinant PKN1 (800 ng) and myelin basic protein (MBP, 200 ng), both from life technologies, were incubated in 50 µL kinase assay buffer (40 mM Tris pH 7.5, 40 mM MgCl_2_, 0.2 mM N-2-hydroxyethylpiperazine-N-2-ethanesulfonic acid pH 7.4, 0.2 mM dithiothreitol, 0.0002% Triton X-100, 0.3 µg/mL BSA) containing 1 µM ATP, 2 µCi γ-^32^P-ATP, 1 µM Phorbol-12,13-dibutyrate (PDBu; C_28_H_40_O, MW: 504.62 g·mol^−1^) and 160 µM phosphatidylserine. After 30 min at 30 °C, the reaction was stopped by adding 5xSDS sample buffer (250 mM Tris pH 11.5, 10% SDS, 50% glycerol, and 25% β-mercaptoethanol). PKC412, a staurosporine derivative, which has been shown to inhibit all PKN isoforms [31](CAS No. 120685-11-2, Sigma), was added at 5 µM. Phosphorylation was analyzed by SDS-PAGE and autoradiography.

### 2.11. Liquid Chromatography–Mass Spectrometry (LC–MS)

Relative quantification of PKN1 phosphorylation was analyzed by LC–MS, with and without the PKN1 kinase inhibitor PKC412. Proteins (5 µg) were reduced in 50 µL 100 mM dithiothreitol in ammonium bicarbonate buffer (100 mM pH 8.0), followed by incubation at 56 °C for 30 min. Proteins were digested with 0.1 µg of trypsin (sequencing grade, P/N: V5111, Promega, Madison, WI, USA) for 6 h at 37 °C under agitation. Free cysteines were alkylated by adding 50 µL 55 mM iodoacetamide to an ammonium bicarbonate buffer, followed by agitation at room temperature for 20 min in the dark. Peptide digests were analyzed using an UltiMate 3000 nano-HPLC system coupled to a Q Exactive Plus mass spectrometer (Thermo Scientific, Bremen, Germany) as described previously [32]. In brief, peptides were separated on a homemade column (100 μm i.d. × 17 cm length) packed with 2.4 μm C18 material (Reprosil, Dr. A. Maisch HPLC GmbH, Ammerbuch, Germany). Solvents for nano-HPLC were 0.1% formic acid (solvent A) and 0.1% formic acid in 85% acetonitrile (solvent B). Total gradient time was 80 min at a flow rate of 300 nL/min. The 20 most abundant peptides in the survey scan were selected for MS fragmentation. The isolation window was set to 1.6 *m*/*z*. Survey full scan MS spectra were acquired from 300 to 1750 *m*/*z* at a resolution of 60,000. Peptides were fragmented by HCD with a normalized collision energy set to 28 and measured in an Orbitrap analyzer at a resolution of 30,000. Data Analysis was performed using Proteome Discoverer 3.0 (Thermo Scientific) with the search engine Sequest HT. Raw files were searched against the uniprot human reference proteome database (last modified 20 July 2023). Precursor and fragment mass tolerance was set to 10 ppm and 0.02 Da, respectively, and up to two missed cleavages were allowed. Fixed modification was carbamidomethyl on C, and variable modifications were oxidation of M phosphorylation on S, T, Y, and acetylation and/or methionine loss of the protein N-terminus. Quantification of phosphorylation rates is based on abundance areas of extracted ions of phosphopeptides and their unmodified counterparts. Ion extraction was performed via Qual Browser 4.2.47 (Thermo Scientific).

### 2.12. Generation of a Polyclonal Antibody against the Phosphorylated PKN1S374

In order to best define the biological significance and kinetics of the PKN1-mediated candidate phosphorylation during neuronal survival/cell death signaling pathways, we generated a pS374-specific polyclonal antibody in rabbits with the Phosphorylation-Specific Antibody Production Services provided by GenScript USA Inc. (Piscataway, NJ, USA) and purified employing the (p)peptide antigen, as the peptide antigen GLYSRSG{pS}LSGRSSLC was used.

### 2.13. Co-Immunoprecipitation of PKN1-AKT

Jurkat human lymphoblastoid CD4+ T cells were cultured in RPMI, supplemented with 1% L-Glutamine, 10% FCS (Fetal Calf Serum) (Sigma, St. Louis, MO, USA), and 1% Penicillin/Streptomycin. The cells were passaged 24 h prior to transfection to yield 80% confluence on the day of transfection. For each transfection, 1 × 10^7^ viable cells were mixed with 15 µg of each plasmid (see below) by electroporation. The plasmids used were pEF HA (empty AKT vector), HA-tagged *h*PKN1 wt, HA-tagged *h*PKN1 S374A, pEF HA-tagged AKT vectors, and HA-tagged *h*PKN1 T774A. After 24 h, the transfected cells were pelleted in ice-cold PBS (8 min, 1200 rpm, 4 °C). The pellets were resuspended in lysis buffer (50 mM HEPES pH 7.5, 2% NP-40, 5 mM EDTA, 5 mM sodium pyrophosphate, 5 mM sodium fluoride, 50 mM sodium chloride, 5 mM activated sodium orthovanadate, 30 µg/mL aprotinin, 30 µg/mL leupeptin) and kept on ice for 15 min. The lysates were then centrifuged at (15 min, 13,200 rpm, 4 °C) and the supernatant was collected. Twenty-five percent of the lysates was added to 4× SDS-PAGE sample buffer as an input control for further SDS-PAGE analysis. For pre-clearing, the lysates (300 µL) were added to 30 µL of Protein-G-Sepharose 4 Fast Flow resin (Cytiva, Marlborough, MA, USA) for 60 min on an end-over-end rotator in a cold room. After pre-clearing, the lysate-beads complex was centrifuged (1 min, 10,000 rpm at 4 °C), and the beads were discarded. Each of the clarified lysates (~300 µL) was then gently mixed with 2 µg of AKT antibody (610860, BD Transduction, Franklin Lakes, NJ, USA) or MOPC-21 antibody (BE0083, Bioxcell, Missouri, TX, USA) for 90 min on an end-over-end rotator in a cold room. The protein–antibody complex was collected and transferred into new tubes containing 30 µL of Protein-G-Sepharose. The protein–antibody–bead complex was obtained by centrifugation (1 min, 10,000 rpm at 4 °C), and the supernatant was discarded. The protein–antibody–bead was washed four times with 500 µL wash buffer (lysis buffer described above, with half the concentration of the protease inhibitors). Finally, 30 µL of elution buffer (containing 60% lysis buffer and 40% 4× Laemmli SDS sample buffer) was added to elute the protein–protein complex, and the mixture was heated for 5 min at 95 °C. The protein–antibody complex was further analyzed by immunoblotting.

### 2.14. Analysis of Colocalization in Immunostained Jurkat Cells

Image correlation analysis was conducted in confocal sections of Jurkat cells stained with AKT and HA using the Colocalization Finder plugin (Image J, version 1.54g) as per online instructions from Philippe Carl (Author). In total, 4–5 cells have been analyzed.

### 2.15. Statistical Analysis

All data are presented as individual *n*-values with or as mean ± S.E.M. For all Cgc culture experiments, *n*-values refer to Cgc prepared from different litters. For comparison of two independent groups, a two-tailed unpaired *t*-test was used. For comparison of three or more groups, a one-way ANOVA with Bonferroni’s or Dunnett’s multiple comparison test was used. All analyses were performed in Graphpad prism 9.

## 3. Results

### 3.1. Pkn1^−/−^ Neurons Have Enhanced AKT Activity and Cell Survival after In Vitro HI

Cgc were exposed to HI (3 h OGD/1 h Rep) and analyzed for the phosphorylation of proteins involved in the AKT pro-survival signaling cascade (Figure 1a, Table 1). WT Cgc showed a reduction in AKT-mediated signaling after HI (Figure 1a–f), with a steep increase in caspase-3 activation (Figure 1f). Interestingly, *Pkn1^−/−^* Cgc showed higher AKT phosphorylation upon HI (Figure 1a–c), which was effectively translated into enhanced phosphorylation of the downstream targets ribosomal protein S6 kinase (p70S6Kinase, Figure 1a,d), glycogen synthase kinase 3β (GSK3β, Figure 1a,e) as well as reduced cleavage of caspase-3 (Figure 1a,f). Post-ischemic AKT hyperactivation upon *Pkn1* knockout was not seen after 24 h reperfusion, suggesting that PKN1-mediated AKT inhibition is particularly relevant during early reperfusion. However, the protective effect of *Pkn1* knockout on caspase-3 activation was seen during longer reperfusion periods (3 h OGD, 24 h Rep, Figure 1g), showing that AKT hyperactivation is effectively translated into improved long-term cell viability. This demonstrates that PKN1-mediated AKT inhibition is relevant during pathophysiological developmental conditions, such as HI.

Reintroduction of human PKN1 (*h*PKN1) into *Pkn1^−/−^* Cgc resulted in a reduction of AKT phosphorylation after HI (Figure 1h). In agreement with our hypothesis that PKN1 exerts a negative effect on post-ischemic cell viability, *h*PKN1-transfected Cgc showed a punctate distribution of PKN1 after HI, which coincided with apoptotic nuclei (Figure 1h,i).

### 3.2. Pkn1^−/−^ Cgc Show Enhanced Axon Outgrowth on Chondroitin Sulphate Proteoglycans

We previously found that *Pkn1^−/−^* Cgc showed enhanced axonal outgrowth on the axon growth-promoting substrate laminin, an effect that was also mediated by over-activated AKT signaling [18]. Since HIE brain injury is characterized by damage with glial scarring [6,7], we chose to analyze the effect of *Pkn1* knockout on axon outgrowth on chondroitin sulfate proteoglycans (CSPGs), substrates of the glial scar that prevent regeneration attempts of axons [8]. We found that *Pkn1^−/−^* Cgc did indeed have an enhanced axonal outgrowth capacity on CSPG (1–10 µg/mL) or CSPG/Laminin, as evidenced by an increase in axon length (Figure 2a,b) and percentage of neurite bearing cells (Figure 2a,c). This suggests that in addition to a protective effect on cell survival, inhibition of PKN1 is also beneficial for axon outgrowth in a growth-inhibitory environment.

### 3.3. Analysis of Functionally Relevant PKN1 Phosphorylation Sites

We next aimed to determine functionally important PKN1 phosphorylation sites. PKN1 has three N-terminal HR1 domains (HR1a-c), a C2-like domain and a C-terminal serine/threonine kinase domain. PKN1 is maintained in an inactive state by dimerization, thereby preventing the interaction of the PKN1 hydrophobic motif with the PDK1 interacting fragment pocket. Recruitment to the cell membrane via the C2-like domain can facilitate lipid binding (such as arachidonic acid) and results in disruption of the dimer. This allows PDK1 to phosphorylate the activation loop in the kinase domain (T774 in human PKN1). Once PDK1 is released, RhoA binds to the HR1a domain and releases pseudo-substrate autoinhibition. Only then PKN1 autophosphorylation can fully activate the molecule [15,16].

Alternatively, PKN1 can be activated by caspase-3-mediated cleavage. This results in a constitutively active protein product (55 kDa) lacking the regulatory N-terminus [33]. This form of deregulated PKN1 activation occurs during apoptosis [34] and has been associated with several pathological brain conditions [35,36,37,38].

Historically, PKN1 activity has been analyzed by kinase assays (Figure 3a) by measuring the phosphorylation of T774 (PDK1 site) or by measuring the proteolytic caspase cleavage product. However, we neither saw an induction of a 55 kDa band nor significant changes in T774 phosphorylation in Cgc. The phosphorylation of the activation loop of PKC family members (including PKN1) is an essential step in their maturation, and accordingly, mutant PKN1 T774A completely abolishes its kinase activity [39]. However, as detailed above, phosphorylation by PDK1 precedes full activity of these enzymes, which is only achieved by autophosphorylation. Therefore, the phosphorylation status of the activation loop itself is generally not considered to be a good indicator of the activity of PKC kinases [40]. We, therefore, next analyzed key phosphorylation sites that could be used to better assess PKN1 kinase activity by relative quantification of PKN1 phosphorylation with LC–MS analysis. First, an in vitro kinase assay was established to analyze if the phosphorylation of the recombinant PKN1 protein and its substrate myelin basic protein (MBP) is inhibited by the PKN1 inhibitor PKC412 [31]. PKN1-dependent autophosphorylation and MBP transphosphorylation were measured by incorporation of inorganic phosphate ^32^Pi from γ-^32^P-ATP, and phosphorylation was analyzed by SDS-PAGE and autoradiography (Figure 3a). Importantly, both PKN1 and MBP phosphorylation were significantly abrogated by the addition of 5 µM PKC412 (Figure 3a).

A few phosphorylation sites with potential regulatory importance have been reported for PKN1 [15,41,42]. Interestingly, T64, S377, and T778 are evolutionarily conserved between mouse, rat, and human PKN1. All candidate phosphorylation sites of PKN1 are listed (http://www.phosphosite.org/proteinAction.do?id=650&showAllSites=true; accessed on 25 October 2023).

Our analysis identified two phosphorylation sites that are inhibited by ~50% upon addition of PKC412, namely T64 and S374 (Figure 3b). Of these two sites, T64 (and its neighbor T63) within the HR1a RhoA-binding domain (Figure 3b) is a potential hit for a major PKN1 autophosphorylation site (already detected by (p)OMICs [42]). S374 has been suggested and validated as a phosphorylation site, which is important for the subcellular localization of PKN1 [43]. 

### 3.4. S374 and T774 PKN1 Phosphosites Have an Inhibitory Effect on Axonal Outgrowth

We next analyzed the functional importance of TT63/64, S374, and T774 for the inhibitory effect of PKN1 on axonal outgrowth by generating phospho-specific alanine mutant-PKN1 constructs and transfecting them into *Pkn1^−/−^* Cgc (Figure 4). S374 is an important regulatory phosphorylation site for PKN1 membrane translocation [42,44] and Rho signaling [43]. We have previously shown that the stimulatory effect of *Pkn1* knockout on axonal outgrowth in Cgc is mediated via aberrant AKT signaling [18]. Therefore, we next tested how the PKN1 alanine exchange mutants TT63/64AA and S374A affect hyperactivated AKT-mediated axonal outgrowth on laminin in *Pkn1^−/−^* Cgc. As a positive control, the kinase-inactive T774A activation loop mutant [18], which has been shown to lose its catalytic activity in vitro [45], was used. We confirmed similar expression levels of all PKN1 mutants (see Figure 4a).

Reintroduction of wildtype (wt) *h*PKN1 into *Pkn1^−/−^* Cgc resulted in a significant reduction (>60%) of axonal length compared to untransfected *Pkn1^−/−^* Cgc (Figure 4b). The neutral exchange TT63/64AA mutant, did not alter this effect. This suggests that the pT63/64 site is functionally not relevant for the inhibitory effect of PKN1 on AKT signaling and, subsequently, axonal outgrowth. As expected, the T774A mutant showed significantly longer axons compared to *h*PKN1 wt. Remarkably, the S374A mutant had a similar effect. Therefore, the pS374 site is of major relevance for the PKN1-mediated inhibitory effect on AKT signaling and axonal outgrowth.

### 3.5. Phosphorylation Levels of PKN1S374 Decrease during Postnatal Cerebellar Development

We next aimed to validate the importance of the phosphorylation status at S374 of PKN1 in postnatal development. We, therefore, designed a phosphosite-specific antibody as a key assessment tool to perform in situ activity measurement of this kinase. The specificity of the novel phospho-specific pS374 antibody was confirmed in Jurkat cells transfected with *h*PKN1 wt and *h*PKN1 S374A. Immunoprecipitation with the pS374 antibody showed a clear reduction in precipitated PKN1 upon transfection with *h*PKN1 S374A (Figure 5a).

We next analyzed cerebellar extracts prepared from P1 to P8 animals for p-PKN1 (S374). Interestingly, phosphorylation of PKN1 at pS374 decreases during development (Figure 5b). This suggests that in addition to the inverse correlation of PKN1 expression and AKT phosphorylation [18], PKN1 S374 phosphorylation itself is also reduced during development.

### 3.6. Role of PKN1S374 in PKN1:AKT Complex Formation

We next tested if the S374 site of PKN1 is mechanistically important for the PKN1:AKT interaction. We transfected Jurkat cells with an empty vector, HA-tagged AKT and HA-tagged *h*PKN1 wt or *h*PKN1 S374A. Co-immunoprecipitation analysis was performed in parallel with an AKT antibody or a mock control antibody (MOPC-21). We found that PKN1 specifically co-immunoprecipitates with AKT but not the mock controls (Figure 6a). Remarkably, the precipitated PKN1 levels were reduced by ~50% upon transfection with the S374A mutant, supporting our hypothesis of an imminent role of the S374 phosphosite for PKN1:AKT interaction (Figure 6a). To further validate this finding, we stained Jurkat cells transfected with *h*PKN1 wt or *h*PKN1 S374A for AKT. In agreement with our hypothesis, we found that the colocalization of PKN1 and AKT was reduced in cells transfected with the PKN1S374 mutant (Figure 6b).

## 4. Discussion

Here, we show that the AKT signaling cascade (p70S6K, GSK3β, caspase-3) is severely inhibited in an in vitro model of HI. This confirms previous findings, showing a strong inhibition of AKT and GSK3β with induction of caspase-3 activity in an in vivo rat HIE model [23]. In that context, it is very promising that the knockout of *Pkn1* results in a profound enhancement of AKT signaling and a significant reduction of caspase-3 mediated apoptosis after HI.

In addition to a detrimental effect on cell viability, another hallmark and potential therapeutic target of HIE is excessive reactive gliosis, which leads to glial scar formation and impedes axon regeneration [46,47]. We have previously shown that PKN1 regulates axonal outgrowth via the inhibition of AKT signaling. Here, we show that the beneficial effect of *Pkn1* knockout on axonal outgrowth is also observed when Cgc are grown on growth inhibitory substrates of the glial scar (CSPGs). Approaches that target both neuronal survival as well as axonal regeneration on inhibitory substrates, are promising tools for enhancing functional recovery after HIE. Therefore, our results validate PKN1 as a novel target molecule for HIE treatment.

However, in order to generate specific inhibitors for this molecule, more information regarding functionally relevant phosphorylation sites is needed.

Our analysis of the enriched (p)peptides by LC–MS identified only a few major PKN1 phosphorylation sites. Among these, the previously described sites TT63/64 and S374 showed the highest decrease in phosphorylation upon incubation with a PKN1 inhibitor. It was shown that T64 phosphorylation in PKN1 is suppressed by cardiolipin [42], and S374 is important for its membrane integration [43]. The conservation of the S374 phosphorylation site in all PKN isoforms further suggests a regulatory role of this site. An interesting possibility is that this site is transphosphorylated, either by PKN1 itself or by some other kinase(s), that may contribute to the regulation of this enzyme [42].

Based on these findings, we tested the effect of various PKN1 phosphorylation mutants on axonal outgrowth of *Pkn1^−/−^* Cgc. Whereas *h*PKN1 TT63/64AA did not alter axon length compared to *h*PKN1 wt, T774A, and importantly, S374A clearly showed that these sites are critical for axon outgrowth and consequently PKN1-AKT interaction (Figure 4). The generation of a novel pS374 antibody allowed us to test the in vivo relevance of this site during development. We have previously shown that PKN1 expression decreased during postnatal development in the cerebellum [18]. Interestingly, we observed a similar trend of p-PKN1 (S374), with a steep decline during early postnatal development. This suggests that in vivo AKT phosphorylation is tightly regulated by PKN1 via both its expression and phosphorylation levels.

To further confirm that S374 is an important phosphorylation site for an AKT-PKN1 interaction, we performed co-immunoprecipitation with AKT, *h*PKN1 wt or *h*PKN1 S374A mutant proteins. Although we still detected an interaction between AKT and *h*PKN1 S374A, it was markedly diminished when compared to *h*PKN1 wt. While the underlying mechanism is not yet clear, our results show that autophosphorylation on PKN1 S374 plays a functional role in PKN1-mediated AKT inhibition.

## 5. Conclusions

In summary, our data shed light on the neurodegenerative effect of PKN1 in neonatal HIE. We validate PKN1 as a promising new therapeutic target, which affects both neuronal survival and axonal outgrowth after in vitro HI. We further established the importance of PKN1 S374 for this effect. These preclinical validation data are exciting findings as, despite the current lack of specific PKN1 inhibitors, the development of such is a focus of cancer research [48] and could benefit future research into neuroprotection and neuroregeneration.

## Figures and Tables

**Figure 1 biomolecules-13-01599-f001:**
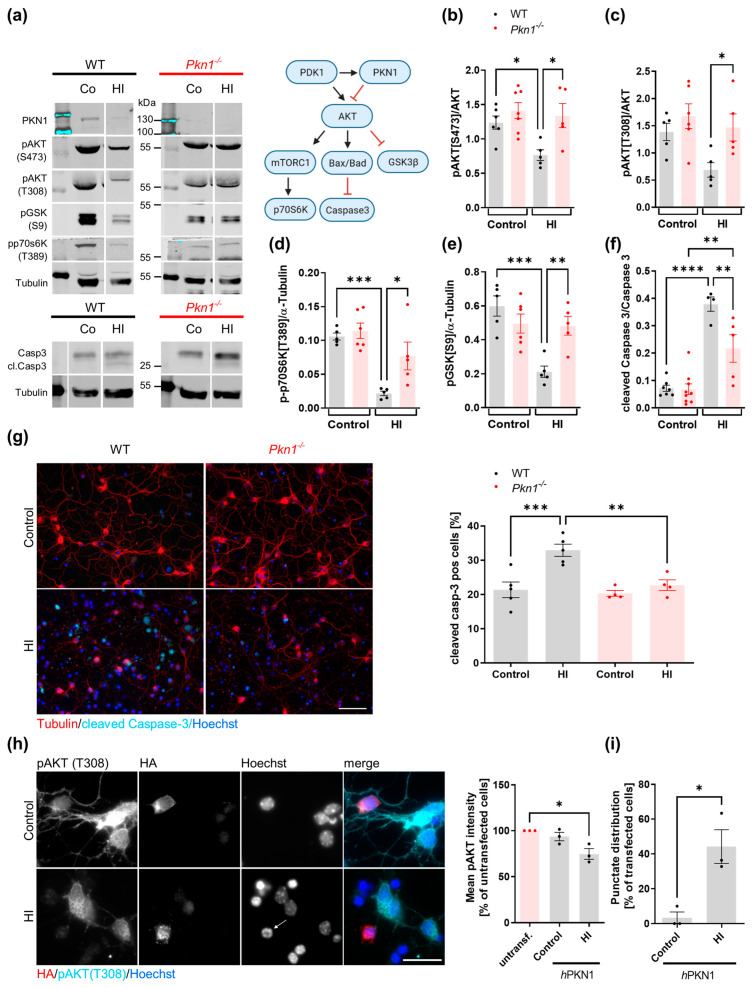
*Pkn1^−/−^* Cgc have enhanced activity of the AKT pro survival-signaling cascade after HI. (**a**) Protein extracts were prepared from Cgc exposed to HI (3 h OGD/1 h Rep), immunoblotted (see Appendix A for uncropped blots), and analyzed for (**a**,**b**) pAKTS473 (one-way ANOVA with Bonferroni’s multiple comparison test, F (3,19) = 5.467, *p* = 0.007, post-test (*) *p* < 0.05), (**a**,**c**) pAKTT308 (one-way ANOVA with Bonferroni’s multiple comparison test, F (3,17) = 4.483, *p* = 0.0171, post-test (*) *p* < 0.05), (**a**,**d**) p70S6KT389 (one-way ANOVA with Bonferroni’s multiple comparison test, F (3,17) = 11.92, *p* = 0.0002, post-test (*) *p* < 0.05. (***) *p* < 0.001), (**a**,**e**) pGSK3BS9 (one-way ANOVA with Bonferroni’s multiple comparison test, F (3,17) = 9.379, *p* = 0.0007, post-test (**) *p* < 0.01, (***) *p* < 0.001) and (**a**,**f**) cleaved and total caspase-3 (one-way ANOVA with Bonferroni’s multiple comparison test, F (3,21) = 25.63, *p* < 0.0001, post-test (**) *p* < 0.01, (****) *p* < 0.0001). (**g**) Cgc were exposed to HI (3 h OGD/24 h Rep) and the percentage of cleaved caspase-3-positive cells in relation to Hoechst-positive cells was counted (one-way ANOVA with Bonferroni’s multiple comparison test, F (3,14) = 11.09, *p* = 0.0005, post-test (**) *p* < 0.01, (***) *p* < 0.001). A total of 34–184 cells per treatment and experiment were analyzed in a blinded manner. Scale bar refers to 50 µM. (**h**) The pAKTT308 intensity of *Pkn1^−/−^* Cgc transfected with HA-tagged *h*PKN1 (*h*PKN1) was reduced after HI (3 h OGD/1 h Rep, data is expressed as percentage of untransfected cells under control and HI conditions, one-way ANOVA with Dunnett’s multiple comparison test, F (2,6) = 9.449, *p* = 0.0140, post-test (*) *p* < 0.05). Scale bar refers to 20 µM. (**i**) *Pkn1^−/−^* Cgc transfected with *h*PKN1 showed a punctate distribution of HA-stained *h*PKN1 after HI (3 h OGD/1 h Rep, two-tailed unpaired *t*-test *t* (4) = 3.996, (*) *p* = 0.0162). A total of 7–28 cells per treatment and experiment were analyzed. Arrowhead points to apoptotic nucleus. All data are presented as individual *n*-values, referring to Cgc preparations from different litters, with mean ± S.E.M. experimenters not blinded to the genotype and treatment except for (**g**).

**Figure 2 biomolecules-13-01599-f002:**
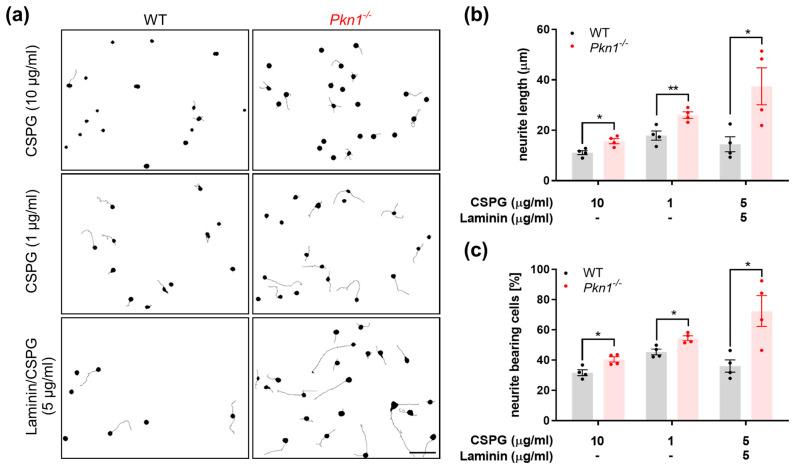
*Pkn1^−/−^* Cgc show enhanced axonal outgrowth on CSPGs, substrates of the glial scar. (**a**) Cgc were grown on CSPG or CSPG/Laminin at indicated concentrations for 1 DIV. Scale bar refers to 50 µM. Cells were stained for TAU and analyzed for (**b**) axon length (CSPG 10 µg/mL: two-tailed unpaired *t*-test *t* (3.5) = 6, (*) *p* = 0.0124, CSPG 1 µg/mL: two-tailed unpaired *t*-test *t* (3.7) = 6, (**) *p* = 0.0099, CSPG/Laminin 5 µg/mL: two-tailed unpaired *t-*test *t* (2.9) = 6, (*) *p* = 0.0268) and the (**c**) percentage of neurite bearing cells length (CSPG 10 µg/mL: two-tailed unpaired *t*-test *t* (3.385) = 6, (*) *p* = 0.0148, CSPG 1 µg/mL: two-tailed unpaired *t*-test *t* (3.683) = 6, (*) *p* = 0.0103, CSPG/Laminin 5 µg/mL: two-tailed unpaired *t*-test *t* (3.3) = 6, (*) *p* = 0.0160). All data are presented as individual *n*-values, referring to Cgc preparations from different litters, with mean ± S.E.M. experimenters not blinded to the genotype or treatment.

**Figure 3 biomolecules-13-01599-f003:**
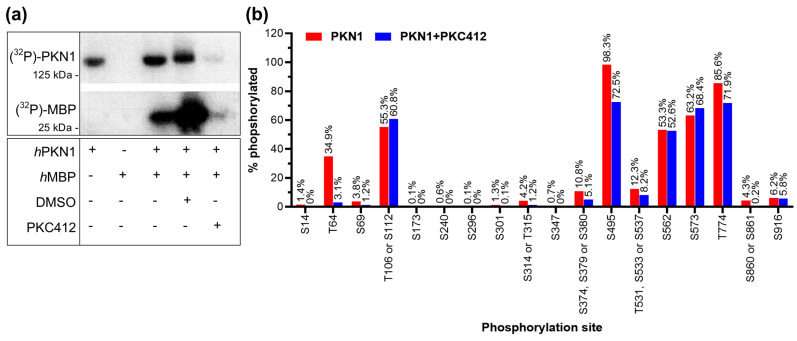
Autophosphorylation of PKN1. (**a**) In vitro protein kinase assay of recombinant human (h)PKN1 and hMBP with or without PKC412 or vehicle (DMSO). The blot is representative of three separate experiments. (see Appendix A for uncropped blots). (**b**) Relative quantification of PKN1 phosphorylation as analyzed by LC–MS with and without the PKC412.

**Figure 4 biomolecules-13-01599-f004:**
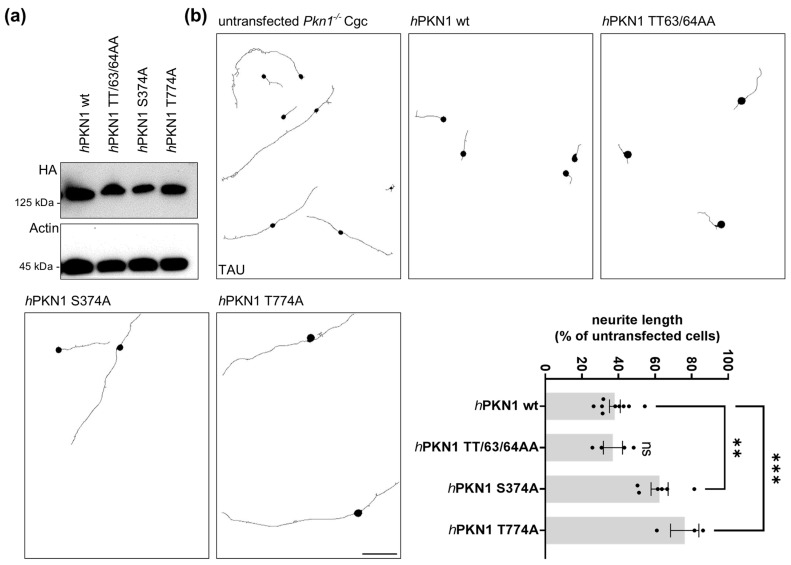
Axonal outgrowth of Cgc is regulated by S374 and T774 PKN1 phosphosites. (**a**) Expression levels of *h*PKN1 wt, *h*PKN1 TT63/4AA, *h*PKN1 S374A, and *h*PKN1 T774A were analyzed by western blotting of protein extracts prepared from Jurkat cells transfected with the different mutants. The blot is representative of three separate experiments (see Appendix A for uncropped blots). (**b**) *Pkn1^−/−^* Cgc were transfected *h*PKN1 wt, *h*PKN1 TT63/4AA, *h*PKN1 S374A or *h*PKN1 T774A and analyzed for axon length at DIV1. For each experiment, the axonal length of transfected cells was expressed as a percentage of the untransfected cells counted (one-way ANOVA with Bonferroni’s multiple comparison test, F (3,18) = 15.13, *p* < 0.0001, post-test (**) *p* < 0.01, (***) *p* < 0.001). Data is presented as individual *n*-values, referring to Cgc preparations from different litters, with mean ± S.E.M. experimenters not blinded to the experimental condition. Scale bars refer to 50 µM.

**Figure 5 biomolecules-13-01599-f005:**
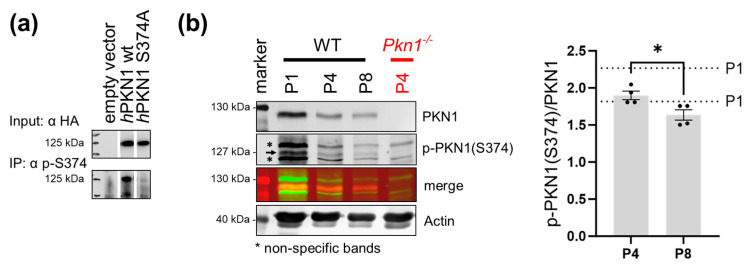
PKN1 phosphorylation at S374 decreases during postnatal development. (**a**) The specificity of the S374 site-specific phosphorylation of PKN1 was confirmed by immunoprecipitation (IP) with the pS374-specific antibody in Jurkat cells, transfected with *h*PKN1 wt or *h*PKN1 S374A. The blot is representative of three independent experiments (see Appendix A for uncropped blots). (**b**) Postnatal whole cerebella lysates (P1 to P8) were analyzed for p-PKN1 (S374) (see Appendix A for uncropped blots, the two separate values for P1 animals are shown as dotted lines in the graph. P4 vs. P8: two-tailed unpaired *t*-test *t* (2.890) = 6, (*) *p* = 0.0277). All data is presented as individual *n*-values referring to different animals with mean ± S.E.M. The black arrow represents the specific p-PKN1 (S374) band, and the stars indicate the non-specific bands of the antibody. Experimenters were not blinded to the experimental condition.

**Figure 6 biomolecules-13-01599-f006:**
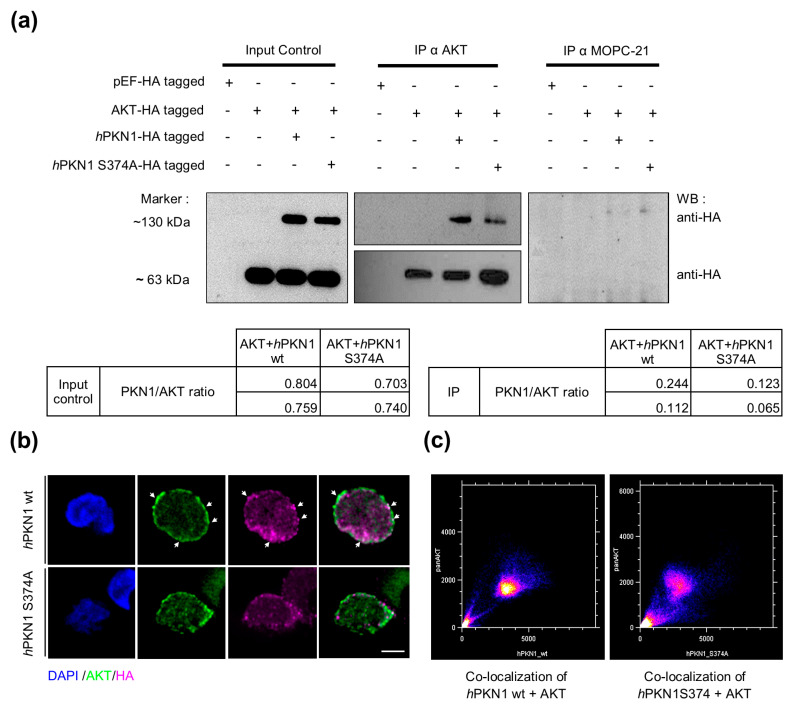
The S374 site plays a fundamental role in the formation of the PKN1:AKT complex. (**a**) Jurkat cells were transfected with the following constructs: pEF/HA (empty AKT vector), pEF HA-tagged AKT, and pEF HA-tagged AKT with HA-tagged *h*PKN1 wt, or HA-tagged *h*PKN1S374A. Co-IP was performed with either an AKT or a mock-control antibody (MOPC-21). Representative western blots probed for anti-HA are displayed. Blots are representative of two independent experiments (see Appendix A for uncropped blots). PKN1/AKT ratios were calculated from the respective HA-bands for input controls and IPs and are shown as tables below the blots for both separate experiments. (**b**) Jurkat cells were transfected with HA-tagged *h*PKN1 wt or HA-tagged *h*PKN1 S374A, and immunostained for AKT and HA. Arrows point to areas with colocalization of AKT and PKN1. (**c**) Image correlation analysis using the Colocalization Finder plugin (Image J) indicates that AKT co-localizes with *h*PKN1 wt (Pearson’s correlation coefficient = 0.929, R = 0.932, Mander’s overlap coefficient M1 = 0.952 and Mander’s overlap coefficient M2 = 0.997) to a greater extent than with *h*PKN1 S374A (Pearson’s correlation coefficient = 0.865, R = 0.870, Mander’s overlap coefficient M1 = 0.944 and Mander’s overlap coefficient M2 = 0.991). Scale bar refers to 15 µm. In total, 4–5 cells have been analyzed. Images are representative of one experiment.

**Table 1 biomolecules-13-01599-t001:** List of all antibodies and respective dilutions used in this study.

Antibody	Company (Catalogue Number)	Application	Dilution/Concentration
pAKT T308	Sigma (SAB4300043)	IF WB	1:100 1:1000
Actin	Sigma (MAB1501)	WB	1:5000
pAKT S473	Cell Signaling (4060)	WB	1:1000
cleaved caspase-3	Cell Signaling (9664)	IF	1:300
caspase-3	Cell Signaling (9665)	WB	1:1000
HA	Cell Signaling (3724)	WB	1:1000
pp70-S6Kinase	Cell Signaling (9204)	WB	1:1000
pGSK3β	Cell Signaling (5558)	WB	1:1000
AKT	Cell Signaling (2920)	WB	1:2000
TAU	Cell Signaling (4019)	IF	1:500
α-Tubulin	Abcam (ab80779)	WB	1:1000
HA.11 Clone 16B12	Covance (MMS-101P)	IF	1:500
PKN1	BD Biosciences (610687)	WB	1:1000
AKT	BD Biosciences (610860)	IP	2 µg
PKN1 pS374 antibody	Self-made, see Section 2.12	WB IP	1:1000 2 µg
InVivoMAb mouse IgG1 isotype control (MOPC-21)	MOPC-21 BioXcell (BE0083)	IP	2 µg
Actin	Santa Cruz (47778)	WB (Jurkat cell extracts)	1:1000

## Data Availability

The data presented in this study are available on request from the corresponding authors.

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
