# Peer review of "PKN1 Exerts Neurodegenerative Effects in an In Vitro Model of Cerebellar Hypoxic–Ischemic Encephalopathy via Inhibition of AKT/GSK3β Signaling"

_biomolecules, 2023, doi:10.3390/biom13111599_

Round 1
Reviewer 1 Report
This manuscript by zur Nedden et al. investigates the role of PKN1 in an in vitro model of HI. Through the use of postnatal cerebellar granule neurons from PKN1 wildtype and knockout mice they demonstrate the pathological relevance of the PKN1-AKT interaction in an in vitro HI model. Moreover, they analyze the PKN1 autophosphorylation at S374, showing that it plays a functional role in PKN1-mediated AKT inhibition and interaction. The manuscript proposes PKN1 as a promising new therapeutic target for Hypoxic-ischemic-encephalopathy, although further extensive research is needed.
Therefore, I support the publication of the manuscript after few minor revisions.
1. It will be helpful to include the concentration/dilutions of the antibodies used in the Materials and Methods section 2.9.
2. The authors analyzed the levels of proteins involved in the AKT pro survival signaling cascade after Cgc exposure to a short reperfusion period (3h OGD/1 h Rep) while for a longer reperfusion period (24h Rep) there is only the analysis of caspase-3 activation. Is there a specific reason for this difference in the analysis? After 24h reperfusion would you still see the AKT pathway alteration?
3. In line 273, the authors claim that “hPKN1-transfected Cgc showed a punctate distribution of PKN1 after HI, which coincided with apoptotic nuclei”. Together with the quantification, it might be helpful for the reader to also include some high resolution or at least high magnification images of the apoptotic nuclei to show the distribution of PKN1.
4. In Figure 1g, 1i, 5b, it is necessary to include in the figure legends the average number of cells counted for each independent experiment.
5. The co-IP results concerning the role of PKN1 S374 in PKN1-AKT complex formation (Figure 6) are interesting, however the number of experiments performed is insufficient to fully support the authors conclusion. I suggest performing at least one more experiment (3 independent experiments are usually the minimum acceptable number). Moreover, it would be also nice to make a quantification with statistical analysis and show the reader a bar graph, as shown for other analysis in the text.
6. The authors showed that PKN1 and PKN1 phosphoS374 expression decreased during postnatal development. Therefore, it would be important to include for the different experiments (Cgc isolation, LC-MS) at which postnatal day the animals were used and why.
Author Response
Response to all the comments from Reviewer 1:
Reviewer 1
This manuscript by zur Nedden et al. investigates the role of PKN1 in an in vitro model of HI. Through the use of postnatal cerebellar granule neurons from PKN1 wildtype and knockout mice they demonstrate the pathological relevance of the PKN1-AKT interaction in an in vitro HI model. Moreover, they analyze the PKN1 autophosphorylation at S374, showing that it plays a functional role in PKN1-mediated AKT inhibition and interaction. The manuscript proposes PKN1 as a promising new therapeutic target for Hypoxic-ischemic-encephalopathy, although further extensive research is needed.
Therefore, I support the publication of the manuscript after few minor revisions.
We want to thank Reviewer 1 for acknowledging the novelty, importance and relevance of our findings. We are extremely grateful to Reviewer 1 for the thoughtful and insightful comments, which have helped us to improve the manuscript considerably.
We have carefully revised the manuscript in accordance with the Reviewer’s requests and added new analyses and revised figures. We sincerely hope that the Reviewer will consider these extensive revisions as addressing their previous concerns in full. The blue text passages in the manuscript mark the major changes referring to both Reviewer comments.
Point by Point response to Reviewer 1:
- It will be helpful to include the concentration/dilutions of the antibodies used in the Materials and Methods section 2.9.
Response to point 1:
We apologize for the lack of clarity in the text. We have included a table (see below) detailing concentration/dilutions of the antibodies to the Materials and Methods section 2.9.
|
Antibody |
Company (Catalogue number ) |
Application |
Dilution/concentration |
|
pAKT T308 |
Sigma (SAB4300043) |
IF |
1:100 |
|
Actin |
Sigma (MAB1501) |
WB |
1:5000 |
|
pAKT S473 |
Cell Signaling (4060) |
WB |
1:1000 |
|
cleaved caspase-3 |
Cell Signaling (9664) |
IF |
1:300 |
|
caspase-3 |
Cell Signaling (9665) |
WB |
1:1000 |
|
HA |
Cell Signaling (3724) |
WB |
1:1000 |
|
pp70-S6Kinase |
Cell Signaling (9204) |
WB |
1:1000 |
|
pGSK3β |
Cell Signaling (5558) |
WB |
1:1000 |
|
AKT |
Cell Signaling (2920) |
WB |
1:2000 |
|
TAU |
Cell Signaling (4019) |
IF |
1:500 |
|
α-Tubulin |
Abcam (ab80779) |
WB |
1:1000 |
|
HA.11 Clone 16B12 |
Covance (MMS-101P) |
IF |
1:500 |
|
PKN1 |
BD Biosciences (610687) |
WB |
1:1000 |
|
AKT |
BD Biosciences (610860) |
IP |
2 µg |
|
PKN1 pS374 antibody |
Self made, see section 2.12 |
WB |
1:1000 |
|
InVivoMAb mouse IgG1 isotype control (MOPC-21) |
MOPC-21 BioXcell (BE0083) |
IP |
2 µg |
|
Actin |
Santa Cruz (47778) |
WB (Jurkat cell extracts) |
1:1000 |
- The authors analyzed the levels of proteins involved in the AKT pro survival signaling cascade after Cgc exposure to a short reperfusion period (3h OGD/1 h Rep) while for a longer reperfusion period (24h Rep) there is only the analysis of caspase-3 activation. Is there a specific reason for this difference in the analysis? After 24h reperfusion would you still see the AKT pathway alteration?
Response to point 2:
As requested by the Reviewer we have analyzed AKT phosphorylation after 24 h reperfusion, and we did not detect a difference between both genotypes at that reperfusion time point. We have included this information and relevant text passages were changed the manuscript as follows (page 7):
Post-ischemic AKT hyperactivation upon Pkn1 knockout was not seen after 24 h reperfusion (data not shown), suggesting that PKN1-mediated AKT inhibition is particularly relevant during early reperfusion. However, the protective effect of Pkn1 knockout on caspase-3 activation was seen during longer reperfusion periods (3 h OGD, 24 h Rep, Figure 1g), showing that AKT hyperactivation is effectively translated into improved long term cell viability.
- In line 273, the authors claim that “hPKN1-transfected Cgc showed a punctate distribution of PKN1 after HI, which coincided with apoptotic nuclei”. Together with the quantification,it might be helpful for the reader to also include some high resolution or at least high magnification images of the apoptotic nuclei to show the distribution of PKN1.
Response to point 3:
We apologize for the suboptimal image and have replaced it with a new representative image at higher magnification in Figure 1h.
- In Figure 1g, 1i, 5b, it is necessary to include in the figure legends the average number of cells counted for each independent experiment.
Response to point 4:
We apologize for the lack of clarity. We have now added the analyzed cell numbers to the figure legends for Figure 1g and 1i. We assume that Reviewer 1 referred to Figure 6b instead of 5b, so we have included the analyzed cell numbers in Figure 6b in the method section 2.14, as well as in the figure legend.
- The co-IP results concerning the role of PKN1 S374 in PKN1-AKT complex formation (Figure 6) are interesting, however the number of experiments performed is insufficient to fully support the authors conclusion. I suggest performing at least one more experiment (3 independent experiments are usually the minimum acceptable number). Moreover, it would be also nice to make a quantification with statistical analysis and show the reader a bar graph, as shown for other analysis in the text.
Response to point 5:
Although our Co-IP experiments were only performed twice, the result provided a clear indication of the regulatory role of the S374A mutation on PKN1 for the physical interaction with AKT (Figure 6a). In order to improve clarity, and as rightfully requested by the Reviewer, the quantitative analysis of the Co-IP data in Figure 6a has been included in the revised Figure 6 and main text, as shown below (page 13):
"Remarkably, the precipitated PKN1 levels were reduced by ~50% upon transfection with the S374A mutant, supporting our hypothesis of an imminent role of the S374 phospho-site for PKN1:AKT interaction (Figure 6a)."
Furthermore, by using a PKN1S374A mutant we could clearly show that the PKN1S374 site is important for PKN1-mediated inhibition of axonal outgrowth (Figure 4), an effect that, as we have previously shown, depends on AKT hyperactivation (zur Nedden et al., JCI, 2018). Furthermore, as mentioned in the manuscript, our results are consistent with published data on a critical role of the (p)S374 site in mediating Rho-induced effects (Zhu et al, 2004).
If the editor insists on adding a third Co-IP experiment, we would require a significantly longer time frame for the revision than the 5 days we have been granted.
- The authors showed that PKN1 and PKN1 phosphoS374 expression decreased during postnatal development. Therefore, it would be important to include for the different experiments (Cgc isolation, LC-MS) at which postnatal day the animals were used and why.
Response to point 6:
We apologize for the lack of clarity and have added the requested information to the text in section 2.2. as follows:
“Cgc were prepared from P6-P8 old animals and kept for 6-8 DIV.”
Additional change in Figure 5b:
In order to clearly and transparently show that we have only collected 2 n-values for P1, we have changed the Figure and statistics. We now show the range of the 2 separate P1 values as dotted lines in the graph and show the statistical comparison of P4 vs P8 as a t-test. Respective changes in the Figure legend are marked in blue.
We thank you for your interest in our study and look forward to hearing from you,
Yours sincerely,
Dr. Stephanie zur Nedden
Ao. Univ. Prof. Gabriele Baier-Bitterlich

Reviewer 2 Report
The presented results of studying the role of PKN1 in the development of ischemic disorders of cerebellar granule neurons and in its neuroprotective effects are undoubtedly of interest. The research design is quite logical and a large amount of data was obtained, and it is obvious that many of the results will or may become the beginning of new hypotheses and research. I enjoyed reading this article and believe that the journal Biomolecules will benefit from the publication of this article.
But I have two minor suggestions to the authors:
1. When first writing the term in full and abbreviating it, it is better to imagine a complete match (if the cerebellar granular cells, then generally accepted cgc, if cerebellar granular neurons, then also a commonly used abbreviation cgn)
2. The fact that the use of a model of ischemia of a 6-8 DIV primary cerebellar culture from newborn rat pups is a model of neonatal Hypoxic–Ischemic Encephalopathy is a controversial issue. In the text, the authors show quite reasonably that the data obtained are also applicable and important for neonatal cerebral ischemia. But it is common for CGC 6-8 DIV to be used to simulate various neurodegenerative models, including adults ischemic pathology. Therefore, I would suggest replacing the word neonatal in the title with the word cerebellar, only in title
Author Response
Response to all the comments from Reviewer 2:
Reviewer 2
The presented results of studying the role of PKN1 in the development of ischemic disorders of cerebellar granule neurons and in its neuroprotective effects are undoubtedly of interest. The research design is quite logical and a large amount of data was obtained, and it is obvious that many of the results will or may become the beginning of new hypotheses and research. I enjoyed reading this article and believe that the journal Biomolecules will benefit from the publication of this article.
We want to thank Reviewer 2 for acknowledging the novelty, importance and relevance of our findings. We are extremely grateful to Reviewer 2 for the thoughtful and insightful comments, which have helped us to improve the manuscript considerably.
We have carefully revised the manuscript in accordance with the Reviewer’s requests and added new analyses and revised figures. We sincerely hope that the Reviewer will consider these extensive revisions as addressing their previous concerns in full. The blue text passages in the manuscript mark the major changes referring to both Reviewer comments.
Point by Point response to Reviewer 2:
- When first writing the term in full and abbreviating it, it is better to imagine a complete match (if the cerebellar granular cells, then generally accepted cgc, if cerebellar granular neurons, then also a commonly used abbreviation cgn)
Response to point 1:
We apologize for the lack of clarity. We have changed the term cerebellar granule neurons to the more common term cerebellar granule cells with its abbreviation (Cgc) in the manuscript.
- The fact that the use of a model of ischemia of a 6-8 DIV primary cerebellar culture from newborn rat pups is a model of neonatal Hypoxic–Ischemic Encephalopathy is a controversial issue. In the text, the authors show quite reasonably that the data obtained are also applicable and important for neonatal cerebral ischemia. But it is common for CGC 6-8 DIV to be used to simulate various neurodegenerative models, including adults ischemic pathology. Therefore, I would suggest replacing the word neonatal in the title with the word cerebellar, only in title
Response to point 2:
In agreement with the Reviewer’s suggestion, we have changed the term ‘neonatal’ to ‘cerebellar’ in the title.
Additional change in Figure 5b:
In order to clearly and transparently show that we have only collected 2 n-values for P1, we have changed the Figure and statistics. We now show the range of the 2 separate P1 values as dotted lines in the graph and show the statistical comparison of P4 vs P8 as a t-test. Respective changes in the Figure legend are marked in blue.
We thank you for your interest in our study and look forward to hearing from you,
Yours sincerely,
Dr. Stephanie zur Nedden
Ao. Univ. Prof. Gabriele Baier-Bitterlich
